# Multi-Scale Spectral-Spatial Attention Network for Hyperspectral Image Classification Combining 2D Octave and 3D Convolutional Neural Networks

Lianhui Liang [1], Shaoquan Zhang [2,*], Jun Li [3], Antonio Plaza [4] and Zhi Cui [1]

1   College of Electrical and Information Engineering, Hunan University, Changsha 418002, China
2   Jiangxi Province Key Laboratory of Water Information Cooperative Sensing and Intelligent Processing, School of Information Engineering, Nanchang Institute of Technology, Nanchang 330099, China
3   Hubei Key Laboratory of Intelligent Geo-Information Processing, School of Computer Science, China University of Geosciences, Wuhan 430078, China
4   Hyperspectral Computing Laboratory, Department of Technology of Computers and Communications, University of Extremadura, E-10071 Caceres, Spain
*   Correspondence: zhangshaoquan1@163.com

**Abstract:** Traditional convolutional neural networks (CNNs) can be applied to obtain the spectral-spatial feature information from hyperspectral images (HSIs). However, they often introduce significant redundant spatial feature information. The octave convolution network is frequently utilized instead of traditional CNN to decrease spatial redundant information of the network and extend its receptive field. However, the 3D octave convolution-based approaches may introduce extensive parameters and complicate the network. To solve these issues, we propose a new HSI classification approach with a multi-scale spectral-spatial network-based framework that combines 2D octave and 3D CNNs. Our method, called MOCNN, first utilizes 2D octave convolution and 3D DenseNet branch networks with various convolutional kernel sizes to obtain complex spatial contextual feature information and spectral characteristics, separately. Moreover, the channel and the spectral attention mechanisms are, respectively, applied to these two branch networks to emphasize significant feature regions and certain important spectral bands that comprise discriminative information for the categorization. Furthermore, a sample balancing strategy is applied to address the sample imbalance problem. Expansive experiments are undertaken on four HSI datasets, demonstrating that our MOCNN approach outperforms several other methods for HSI classification, especially in scenarios dominated by limited and imbalanced sample data.

**Keywords:** hyperspectral images (HSIs); deep learning; convolutional neural networks (CNNs); 2D octave convolution; DenseNet

## 1. Introduction

Hyperspectral images (HSIs) are captured by hyperspectral imagers, and consist of tens or even a few hundred consecutive narrow bands of the spectrum. These instruments are capable of collecting abundant spectral and spatial information [1–3]. The resulting data cubes can be used to accurately identify ground materials and discriminate ground covers. Thus, HSIs have been extensively applied in various domains, including environmental monitoring [4], agriculture construction [5], military defense [6], anomaly detection [7], and so on. The task of HSI categorization aims to allocate a unique and concrete category to individual HSI pixels based on their feature information. HSI classification is an essential branch in HSI processing tasks which has already become a research hotspot among remote sensing fields [1–3].

During the past decades, many approaches have been proposed to classify HSIs. Early HSIs classification approaches mainly used spectral features for classification, such as SVM [8], decision trees [9], and multi-nomial logistic regression (MLR) [10]. As is well

known, adjacent pixels of HSI are strongly correlated, which leads to a high probability that they belong to the same ground category. The classification methods merely based on spectral characteristics overlook the relevance of spatial information and the consistency of local feature information, resulting in an insufficient utilization of spatial contextual feature information. Therefore, aiming to further enhance their classification performance, many machine learning-based approaches for HSI classification tasks integrate both spectral and spatial information in their models. For example, Li et al. [11] proposed a spectral-spatial feature categorization method that combines subspace polynomial logistic regression and Markov random field (MRF), based on MLR to learn posterior probability distributions from spectral signatures and the subspace projection-based approach to represent noisy and height-mixing pixels. The spatial contextual information is then included by employing multi-level MRF prior. Notably, MRF usually exploits the continuity of neighboring image pixel labels in the probabilistic sense, and it establishes spatial context dependencies by specifying local conditional probabilities [12,13]. Shen et al. [14] proposed an approach for spectral-spatial feature information extraction by conducting complex 3D Gabor wavelet groups with various frequencies and directions. In [15], a pixel-level and object-level feature extraction method was first proposed by using a composite kernel scheme, which merges the features of these two levels into a composite kernel and feeds them into an SVM for classification, and then the voting fusion scheme is employed to determine the final categorization results. Besides, conditional random fields [16], extended attribute profiles [17], multiple kernels [18], and extinction profiles [19] have also been proposed. However, although the above traditional HSI classification approaches have achieved good categorization results, they are largely based on shallow learning and handcrafted features, relying on professional domain knowledge in the parameter setting stage. It is, therefore, difficult to obtain the deep-level abstract characteristic information contained in HSIs using these approaches.

Recently, with the rapid evolution of deep learning (DL) techniques in machine vision, audio processing, natural language, face recognition, and so on, DL-based techniques have been extensively introduced to remote sensing image processing tasks [1,2,20]. Some typical DL approaches have been widely employed in HSI classification, involving stacked auto-encoders, recurrent neural networks, deep belief networks, and CNNs, which significantly boost the performance of the HSI categorization task [21–25]. DL-based techniques are capable of automatically extracting HSI features from a low to high level and learning efficient and deeper abstract features. Chen et al. [26] exploited 3D CNNs to obtain spectral-spatial joint feature information from the raw data directly. Hang et al. [27] proposed an attention-aided CNN model that merges the attention modules into convolutional layers, making the CNN concentrate on more discriminative channels and spatial locations. Although deeper DL networks are capable of extracting more refined discriminative features, it is more difficult to train a deeper model due to the vanishing gradient phenomenon. Generally speaking, the deeper the DL network is, the easier it is to lead to gradient disappearance or explosion problems. ResNets [28] and DenseNets [29] effectively alleviate the vanishing gradient phenomenon. Based on ResNets and 2D CNNs, Lee et al. [30] utilized multi-scale convolutional filter groups to construct a nine-layer, fully connected CNN to realize the joint extraction of spectral-spatial features. In [31], two 3D CNN dense blocks with different kernel sizes, spatial densely block and spectral densely block, were proposed to acquire deeper spatial and spectral characteristics. Li et al. [32] proposed a deep multi-layer fusion DenseNet model, which utilizes 2D CNN and 3D CNN dense blocks to capture different levels of spatial and spectral signatures from the original HSI, respectively. In [33], a spatial and spectral signature extractor combining multi-scale 2D DenseNet with Bi-RNN is proposed to enhance the acquisition of complex spatial contextual feature information and the propagation of features among various convolutional layers via utilizing multi-scale 2D DenseNet.

Based on 3D CNNs and DenseNet, Ma et al. [34] developed a double branch model (DBMA) combined with 3D DenseNet blocks and multiple attention mechanism blocks,

applying both attention mechanisms to emphasize important spectral and spatial components. Inspired by [34], a double branch and dual attention mechanism framework (DBDA) is developed, employing input patches of various sizes, to refine and spotlight the extracted feature maps by introducing different channel and spatial attention modules [35]. Since substantial spatial redundant information exists in traditional CNN-based networks, Chen et al. [36] first proposed to adopt octave convolution instead, which not only reduced redundant spatial information, but also improved performance through effective communication between high and low frequencies. In order to deeply mine discriminative spectral-spatial features and simultaneously decrease the spatial feature redundancy caused by CNNs, Tang et al. [37] developed a 3D octave convolution (3Doc-conv) model with both spectral and spatial attention networks by replacing the traditional CNN with octave convolution. Xu et al. [38] presented a multi-scale 3Doc-conv with spatial and channel attention to obtain the spatial contextual signatures at different scales via multi-scale octave convolution kernels. Wang et al. [39] designed a fast multi-scale capsule model with octave convolution, reducing the spatial redundancy parameters and obtaining competitive classification performance. Combining the characteristics of 2D CNNs and 3D CNNs, a hybrid spectral CNN model is proposed to use 2D CNN after the 3D CNN to capture more deep spatial representation features, reducing the complexity by using the 3D CNN alone [40]. Moreover, a cross-level spectral-spatial joint coding model was also introduced for the categorization of HSIs, which enhances the weight of small samples with classification difficulties by developing a category-proportional sampling strategy and a weighted loss method [41]. In [42], a model combined with global convolutional long short-term memory and global joint attention mechanism is introduced, utilizing a hierarchical balanced sampling method and a weighted softmax loss strategy to further overcome the problem of insufficient and unbalanced samples in HSIs.

Inspired by the above-mentioned works, here, we develop a new multi-scale spectral-spatial attention network framework combining 2D octave and 3D CNNs (MOCNN) for classification tasks. The spectral-spatial feature extraction module of our MOCNN is composed of two primary branch networks: the multi-scale DenseNet based on 3D CNNs (multi-scale 3D DenseNet) and the multi-scale 2D octave convolution network (multi-scale 2D octave) are, respectively, applied to spectral signatures and spatial contextual feature information extraction. For the spatial feature extraction sub-network, a multi-scale 2D octave is employed to adequately mine the spatial contextual feature information, while decomposing the obtained feature map into low-frequency (LF) and high-frequency (HF) components to decrease the redundancy of spatial feature information. For the spectral feature extraction sub-network, the multi-scale 3D DenseNet is utilized to sufficiently mine the spectral signatures at various scales and simultaneously merge the spectral features in shallow and deep convolutional layers to improve the reuse of spectral characteristics. Additionally, the spectral attention mechanism module is used to allocate proper weight values for each spectral band while suppressing insignificant spectral bands to mitigate the effect of redundant HSI bands in the classification. The channel attention mechanism module is used in the two feature extraction sub-networks to enhance the interactions of the feature map information among feature channels and improve the feature extraction capability. Finally, a sample balancing strategy based on the weighted cross-entropy loss function (WCL) is used to address the problem of sample imbalance. The MOCNN proposed in the present paper is inspired from [43]. The major contributions of this study are outlined below:

- Based on 2D octave, a multi-scale 2D octave convolution sub-network is proposed to capture spatial feature information. It can not only reduce the spatial feature information redundancy, but also extract complex spatial structure information adequately.
- A multi-scale DenseNet based on 3D CNNs is exploited to adequately explore the discriminative spectral signatures at various scales, while fusing the spectral signatures in both shallow and deep convolutional layers to enhance the reuse of spectral features.

- Two types of attention models are utilized to highlight the features containing important information to boost the spectral-spatial feature capture capability of the model. Furthermore, in order to address the problem of sample imbalance, a sample balancing strategy based on the WCL is employed to achieve the balance of the weight probabilities for each category, resulting in the fact that the model focuses more on categories with scarce training samples.

The organization of the remaining parts of this article is described below. The related materials and methods (i.e., the proposed MOCNN model, 2D octave, 3D DenseNet, attention mechanism models, and balanced sampling strategy) are briefly presented in Section 2. The parameter configuration, experimental setup, extensive experimentation and discussion, analysis of experimental results at various sample ratios, and ablation experiments are detailed in Section 3. Ultimately, Section 4 provides conclusions and directions regarding future research.

## 2. Materials and Methods

The proposed MOCNN model primarily consists of three modules: a branch network of the multi-scale 2D octave for spatial contextual feature information extraction, a branch network of the multi-scale 3D DenseNet for exploring the spectral feature information, and a spectral-spatial feature fusion sub-network. The flowchart of the network of the MOCNN approach for HSI classification is illustrated in Figure 1. For spatial feature extraction, after dimensionality reduction of the raw HSI data by principal component analysis (PCA), two 2D octave convolutional networks (where each network contains four 2Doc-conv) are utilized to decompose the feature map into LF and HF components to decrease the redundancy of spatial feature information. Furthermore, the multi-scale 2D octave with two distinct convolution kernel sizes is used to further capture complex spatial contextual feature information of various scale sizes. In multi-scale 3D DenseNet, firstly, a band attention mechanism (BAM) is used to band select and weight each spectral band, to improve the acquiring of characteristic information contained in vital spectral bands and decrease band redundant feature information. Subsequently, the multi-scale 3D DenseNet is exploited to adequately explore the spectral characteristic information at various scales, while fusing the spectral features among various layers in the shallow and deeper convolutional layers. In the final part of the above two sub-networks, the channels of the feature maps connected by concat operation are weighted using the efficient channel attention mechanism (ECA) for enhancing the information capture capability of the spectral and spatial feature channels. Lastly, in the feature fusion sub-network, the final feature maps gained via connecting these two sub-networks are fed through the fully connected layer to yield the new spectral and spatial fusion features. The fused new features are then imported to the softmax layer to gain prediction results.

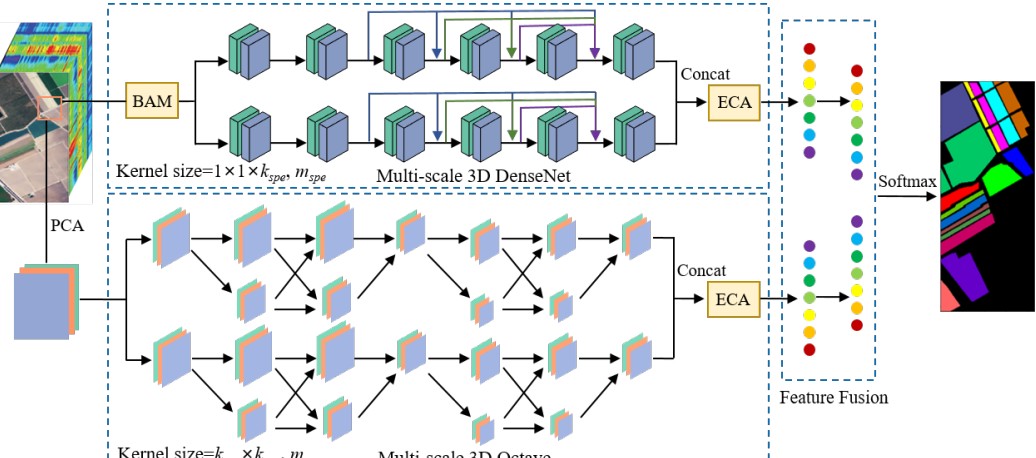

**Figure 1.** The flowchart of the network of the MOCNN approach for HSI classification.

### 2.1. 2D Octave Convolution

The image information is capable of being divided into the LF information component and the HF information component. Among them, the LF information denotes the relatively smooth or homogeneous regions of the image, which have relatively slow image gray value transformations. The HF information refers to the image boundary information and the detail information of small areas of the image, where the image gray value transforms relatively dramatically. Likewise, the output characteristic maps in the CNNs can be considered as the characteristic signal made up of LF and HF information, and each position in the spatial dimension of the characteristic map stores its own feature signal individually. Octave convolution stores the LF information that transforms slowly between adjacent positions together in a low-resolution tensor by sharing positions so as to decrease the resolution of LF features and decrease the superfluity of spatial characteristics and operational complexity. Hence, 2D octave convolution (2Doc-conv) is applied to obtain spatial contextual feature in this work, which can effectively minimize the spatial information superfluity and lower the computational effort of the model. The schematic of the overall structure of 2Doc-conv network is depicted in Figure 2.

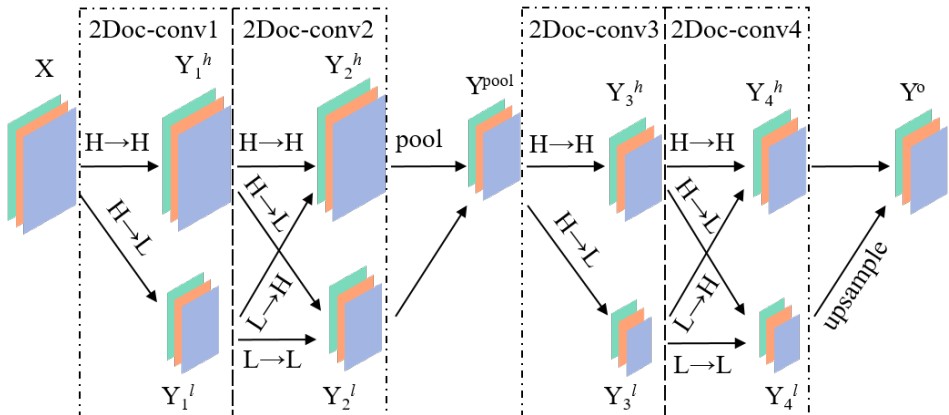

**Figure 2.** The schematic of the overall structure of the 2D octave convolutional network.

Specifically, let us suppose that the input feature maps of the 2Doc-conv are denoted as $X=(X^h, X^l)$, and the output at the $m$-th layer of the octave convolution is denoted by $Y_m=(Y_m^h, Y_m^l)$, where the upper label $h$ and the upper label $l$ stand for the corresponding feature maps belonging to HF and LF information, respectively. Notice that $L \rightarrow L$ and $H \rightarrow H$ denote the exchange of information between intra-frequencies. $H \rightarrow L$ and $L \rightarrow H$ express the exchange of information between HF and LF. As the raw HSI contains abundant spatial and spectral signature detail information, it can be regarded as HF feature data. As displayed in Figure 2, the 2Doc-conv network is made up of four 2Doc-convs, an averaging pooling (AP) operation, and an upsample layer. The output characteristic information of the first layer in this 2Doc-conv is expressed as:

$$Y_1^h = Con(X^h), \tag{1}$$
$$Y_1^l = Con(Apool(X^h)), \tag{2}$$

where $Con(.)$ and $Apool(.)$ represent the 2D convolution and AP operation, respectively. In the second layer of the 2Doc-conv network, it is mainly primarily utilized to complete the update of same-frequency (HF to HF or LF to LF) feature information and the transformation between different-frequency (HF to LF or LF to HF) ones. Thus, the output features of the second layer in this 2Doc-conv can be formulated as:

$$Y_2^h = Con(Y_1^h + Upsample(Y_1^l)), \tag{3}$$
$$Y_2^l = Con(Y_1^l + Con(Apool(Y_1^h))), \tag{4}$$

where $Upsample(.)$ represents the upsample operation based on bi-linear interpolation. For reducing the redundancy of feature information, the HF information acquired in the second layer is downsampled, and then the LF feature information gained in this layer is merged to yield a novel feature $Y^{pool}$, whose calculation formula is:

$$Y^{pool} = Con(Apool(Y_2^h) + Con(Y_2^l)). \tag{5}$$

Likewise, the final output feature information of the third and fourth layer, $Y_3^h$, $Y_3^l$, $Y_4^h$, and $Y_4^l$, in this network can be updated by the communication mechanism between intra-frequency and inter-frequency, respectively. Finally, $Y^o$ can be represented by:

$$Y^o = Con(Con(Y_4^h) + Upsample(Y_4^l)). \tag{6}$$

To sum up, based on 2Doc-conv, the slowly varying LF feature information is stored in the tensor with LF resolution, and then the LF feature resolution is lowered by sharing information among adjacent positions. Compared with traditional CNNs, octave convolution has two obvious advantages, as it can not only reduce spatial redundancy but also extend the receptive field of the network. Therefore, the 2D octave is exploited instead of CNNs to improve the categorization performance of HSI in this work.

### 2.2. 3D DenseNet

Generally, traditional CNN models increase the depth of the network model by stacking convolutional layers, thereby improving the classification accuracy. However, the problem of gradient descent or gradient explosion will occur when too many convolutional layers are stacked. Each convolutional layer of DenseNet gains external input information from all previous convolution layers and delivers their own feature information to all succeeding convolution layers. The gradient value gained in each convolution layer is the gradient addition from the previous convolution layers, which can effectively alleviate the gradient descent problem [29]. Compared with ResNet [28], the DenseNet structure with feature multiplexing can utilize hierarchical features more efficiently and reinforce feature transfer among convolution layers.

All convolutional layers in 3D DenseNet are connected by directly skip-connecting to achieve maximum information transfer among layers of the network. In short, the input of each convolutional layer is the concatenation of the outputs of all previous convolution layers. The first 3D convolutional layer processes raw pixel data of size a × a × B (where a × a denotes the spatial size and B denotes the number of bands) to generate $m_0$ feature maps of size a × a × b. These maps serve as input to dense blocks. As illustrated in Figure 3, the input of the dense block is of the size a × a × b with $m_0$ feature maps, where the subscript indicates the amount of convolutional layers within the dense block. The convolution layer embedded within the dense block is represented as D(.), after each convolution layer applies m kernels of size k × k × c to extract rich spectral information, the output feature map of each layer is size of a × a × b. The number of output spectral feature maps of the $t$-th layer can be formulated as $m_0 + (t-1) \times m$, where $m_0$ means the number of channels contained in the initial feature map. The input to the $t$-th layer of the densely connected model can be expressed as:

$$H_t = D(H_0, H_1, H_2, ......, H_{t-1}) \qquad t \in N^+, \tag{7}$$

where $D$ denotes a function module, which includes convolutional layer, batch normalization (BN), and activation function layer. $N^+$ denotes a positive integer.

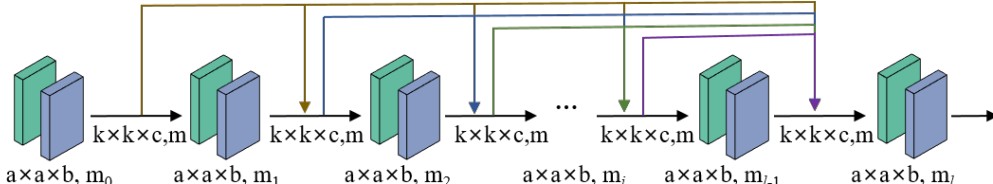

**Figure 3.** Schematic of the structure of 3D DenseNet.

## 2.3. Attention Mechanism Module

The attention mechanism can be used to focus on a certain part of the HSI, assign independent and appropriate weights, weaken redundant and useless information, and highlight effective features that are beneficial for classification. This section will describe the BAM and ECA attention modules in detail.

### 2.3.1. Spectral Attention Mechanism

The bands of hyperspectral images are redundant and noisy, whereas a band attention module has the ability to conduct band selection and weight each band, which can mitigate the negative effects of band superfluity in HSI classification [44]. The schematic of the structure of BAM is shown in Figure 4. The BAM module primarily contains three $3 \times 3$ 2D convolutional layers, two 1D convolutional layers, two pooling layers, and an activation function layer, where the number of channels of 2D convolutional layers is 16, 32, and 32, respectively. In this attention model, after using a 2D convolutional layer for expanding the receptive field, the resolution is reduced by pooling layers to obtain global information in the spatial domain, and then a 1D convolutional layer is employed to further learn the non-linear correlation among the bands. Ultimately, a sigmoid function is adopted to obtain the spectral band weight vector. Notably, the sizes of the two 1D convolution kernels are $1 \times 1 \times 32 \times B/r$ and $1 \times 1 \times B/r \times B$, in which B denotes the number of spectral bands and r is a hyper-parameter that controls the degree of information aggregation layers in the 1D convolution layer. The forward propagation expression of BAM is represented as:

$$U_o = \psi_2(W_{12}\psi_1(W_{11}f_G(\psi_1((W_{23}\psi_1(W_{22}f_P(\psi_1(W_{21}U_i)))))))), \tag{8}$$

where $U_i$ and $U_o$ denote the HSI data input to the BAM and the output of the BAM, $W_{ij}$ denotes the j-th i-dimensional convolutional kernel matrix in the BAM, $f_P$ denotes an AP layer for resolution reduction, and $f_G$ denotes a global AP layer for the sufficient fusion of the spatial feature information contained in the generated feature maps to form a band mask. Moreover, $\psi_1$, $\psi_2$ denote the Relu and sigmoid activation functions. After obtaining the spectral band weighted vector by the $\psi_2$ function, the inner product operation is performed with the input of BAM to obtain more representative spectral band features.

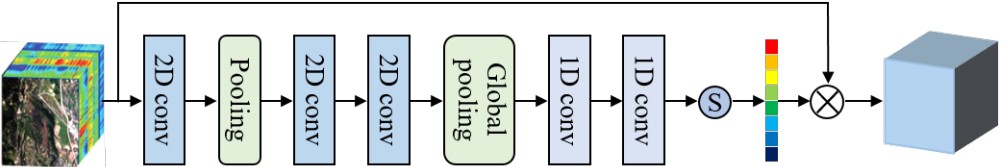

**Figure 4.** The schematic of the structure of BAM. S and $\otimes$ denote the sigmoid activation function and the multiplication of elements, respectively.

### 2.3.2. Channel Attention Mechanism

The proper cross-channel feature interaction and avoidance of dimensionality reduction for channel attention learning can enhance the feature learning capability of the model while lowering its complexity. ECA is a lightweight and local cross-channel feature interaction attention model without dimensionality reduction [45]. The schematic of the structure of ECA is given in Figure 5. It generates channel attention weight parameters via 1D

convolution, with the size R of the convolution kernel being self-adaptively determined by a non-linear mapping of channel dimensions C, which leads ECA to significantly lower the model complexity while sustaining cross-channel interaction. Its computational expression can be presented as:

$$Q_C = P(I_C) = \frac{1}{S \times S} \sum_i^S \sum_j^S I_C(i,j),\tag{9}$$

where $I_C(i,j)$ refers to the characteristic information of the *c*-th channel from the input feature maps at the $(i,j)$ coordinate position. $Q_C$ denotes the feature information of the *c*-th channel from the feature map obtained after the AP layer of $I_C$. The size R of the 1D convolution kernel (the coverage of local cross-channel interactions) is expressed as:

$$R = \alpha(C) = \left| \frac{\log_2(C)}{\beta} + \frac{b}{\beta} \right|_{od},\tag{10}$$

where $|r|_{od}$ represents taking the nearest odd value of *r* and the values of parameters *b* and $\beta$ are given as 1 and 2, respectively. Hence, after the values of parameters *b*, $\beta$ are defined, the value of R is exclusively defined adaptively by the value of channels C. Finally, the corresponding weights of the channels can be calculated as:

$$w_C = \psi_1 \left( \sum_{j=1}^R W^j Q_c^j \right), \qquad Q_c^j \in \theta_c^R,\tag{11}$$

where $\psi_1$ denotes the sigmoid activation functions and $\theta_c^R$ represents the set of *R* adjacency channels of $Q_C$.

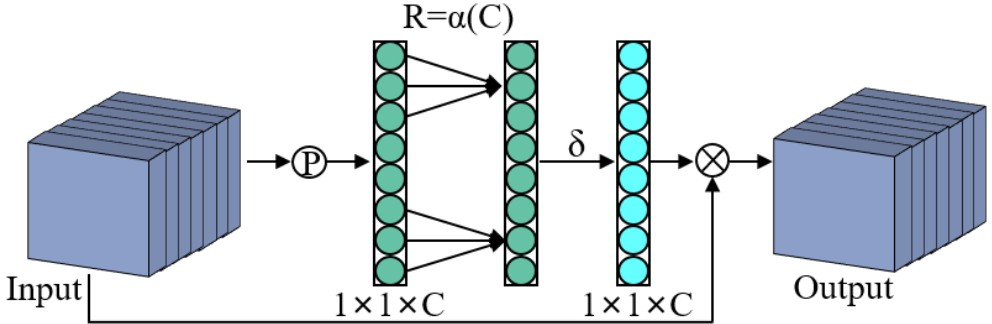

**Figure 5.** The schematic of the structure of ECA. Here, *I* stands for the input feature map of the ECA, *C* represents the number of channels of *I*, *P* denotes the set of features obtained from the input feature map *I* after AP, and $\otimes$ denotes the multiplication of elements.

### 2.4. Balanced Sampling Strategy

The WCL can balance the weight probabilities of different categories, which is a common method to address the imbalance problem [46]. To tackle the problem of sample imbalance in this paper, the WCL is applied to lower the weights of easily classified sample categories, while making the model more focused on categories with small samples and misclassification. Based on this balanced sample strategy, the composite loss function of the MOCNN model can be represented as:

$$L_{im}(y, P(Z_{im})) = -\sum_{i=1}^{n} W_i[ylog(P(Z_{im}) + (1-y)log(1 - P(Z_{im})))], \tag{12}$$

$$L_{spa}(y, P(Z_{spa})) = -\sum_{i=1}^{n} [ylog(P(Z_{spa}) + (1-y)log(1 - P(Z_{spa})))], \tag{13}$$

$$L_{spe}(y, P(Z_{spe})) = -\sum_{i=1}^{n} [ylog(P(Z_{spe}) + (1-y)log(1 - P(Z_{spe})))], \tag{14}$$

$$L = L_{spa}(y, P(Z_{spa}) + L_{spe}(y, P(Z_{spe})) + \eta L_{im}(y, P(Z_{im}))), \tag{15}$$

where $Z_{im}$, $Z_{spa}$, and $Z_{spe}$ denote the logits of the spatial-spectral feature fusion, spatial feature extraction, and spectral feature extraction sub-network in the proposed model, respectively, $L$ denotes the composite loss function of the proposed MOCNN model, $L_{im}$, $L_{spa}$, and $L_{spe}$ denote the loss functions of the corresponding sub-networks, separately, $y$ denotes the true label corresponding to the $i$-th training sample, $\eta$ denotes the parameter that controls the weight of $L_{im}$, which is set empirically to 0.9, and $W_i$ denotes the weight parameter corresponding to the category loss, which can be calculated from:

$$W_i = \frac{N}{T_i}, \tag{16}$$

where $N$ stands for the number of occurrences of the median across all categories and $T_i$ denotes the frequency of category $i$, and it represents the proportion of the number of categories $i$ to the sum of all pixels in the training sample. Thus, $T_i$ is expressed as:

$$T_i = \frac{M_i}{\sum_j M_j}, \tag{17}$$

where $M_i$ refers to the number of samples of category $i$. During each iteration of training, the network loss corresponding to the training set is updated by the counter-propagation operation per layer, and the optimal network is selected in a finite number of iterations.

## 3. Experimental Results and Discussion

### 3.1. Experimental Datasets Description

In this section, to verify the performance of the MOCNN model and contrast it with other approaches, four HSI datasets are applied in the experiments, i.e., the Pavia University (UP), Salinas Valley (SV), Indian Pines (IN), and Zaoyuan (ZY) datasets. The reference maps (i.e., false-color images and ground truth) for the four HSIs are presented in Figure 6. Three common quantitative indicators, namely, overall accuracy (OA), average accuracy (AA), and Kappa coefficient (Ka), are utilized to evaluate the classification performance of all methodologies [47]. In particular, OA means the percentage of correct categorization of all pixels, AA refers to the mean classification precision of the whole category, and Ka takes into account the influence of uncertainty on the classification results and reflects the coherence between classification outcomes and the ground truth. The values of OA, AA, and Ka vary from 0 to 1, where closer to 1 implies better classification outcomes of the model.

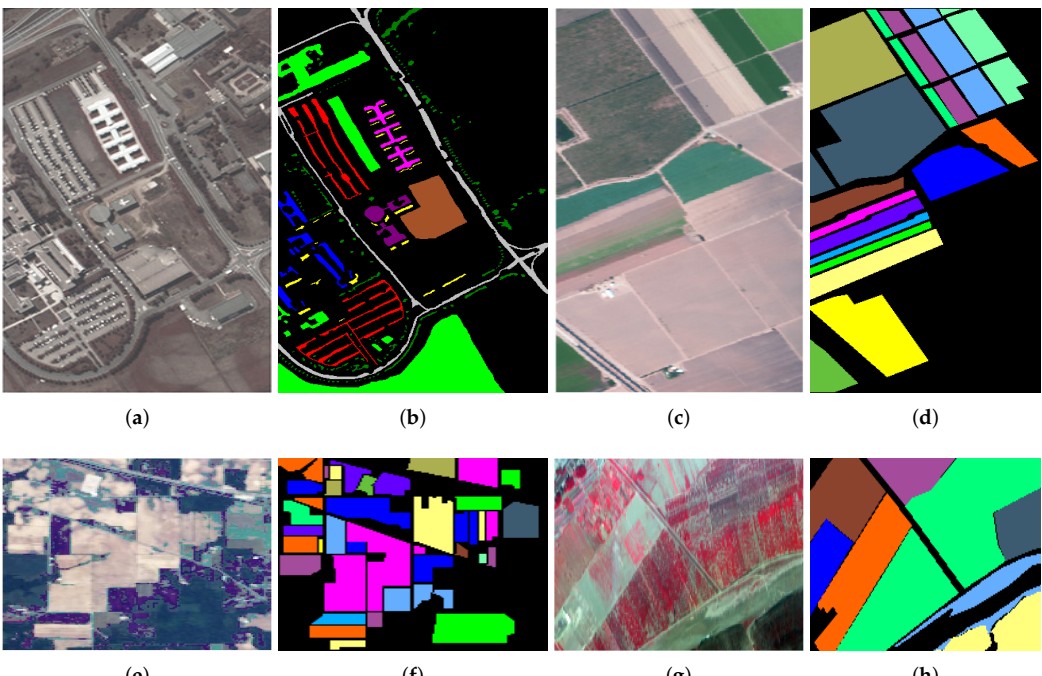

**Figure 6.** Reference maps of the four HSIs. In this figure, sub-maps (**a**,**c**,**e**,**g**) refer to the false-color images of the UP, SV, IN, and ZY, whereas sub-maps (**b**,**d**,**f**,**h**) refer to the ground truth corresponding to them.

**Pavia University dataset (UP):** The UP contains 610 × 340 pixels and encompasses 103 spectral channels after abandoning noisy spectral channels, covering wavelengths varying from 430 to 860 nm, whose spatial resolution (SR) is up to 1.3 m. It encompasses nine classes of labeled samples of land cover objects for classification.

**Salinas Valley dataset (SV):** The SV contains 512 × 217 pixels and encompasses 204 spectral channels after abandoning 20 noisy spectral channels, covering wavelengths varying from 360 to 2500 nm with 3.7 m SR. It encompasses 16 classes of various land cover labeled samples for classification.

**Indian Pines dataset (IN):** The IN contains 145 × 145 pixels and encompasses 220 spectral channels. It covers wavelengths varying from 400 to 2500 nm, while its SR is 20 m. It encompasses a total of 16 classes of land cover labeled samples for classification.

**Zaoyuan dataset (ZY):** The ZY was obtained by OMIS sensor in 2001 in the Zaoyuan zone of China. It encompasses 80 spectral channels after discarding 48 noisy spectral channels and covers wavelengths varying from 400 to 1700 nm. The spatial size of ZY is 137 × 202 with 23,821 labeled pixels. It encompasses a total of eight classes of land cover labeled samples for classification.

During each experimental execution, the selected experimental samples (training samples, validation sets, and test samples) are taken through a random sampling method. Tables 1–4 provide the categories information and corresponding sample numbers for each HSI dataset.

**Table 1.** Category information and corresponding experimental sample size for UP.

| Category | Total | Training | Validation | Test |
|---|---|---|---|---|
| C1 | 6631 | 100 | 100 | 6431 |
| C2 | 18,649 | 280 | 280 | 18,089 |
| C3 | 2099 | 32 | 32 | 2035 |
| C4 | 3064 | 46 | 46 | 2972 |
| C5 | 1345 | 21 | 21 | 1303 |
| C6 | 5029 | 76 | 76 | 4877 |
| C7 | 1330 | 20 | 20 | 1290 |
| C8 | 3682 | 56 | 56 | 3570 |
| C9 | 947 | 15 | 15 | 917 |
| Total | 42,776 | 646 | 646 | 41,484 |

**Table 2.** Category information and corresponding experimental sample size for SV.

| Category | Total | Training | Validation | Test |
|---|---|---|---|---|
| C1 | 2009 | 21 | 21 | 1967 |
| C2 | 3726 | 38 | 38 | 3650 |
| C3 | 1976 | 20 | 20 | 1936 |
| C4 | 1394 | 14 | 14 | 1366 |
| C5 | 2678 | 27 | 27 | 2624 |
| C6 | 3959 | 40 | 40 | 3879 |
| C7 | 3579 | 36 | 36 | 3507 |
| C8 | 11,271 | 113 | 113 | 11,045 |
| C9 | 6203 | 63 | 63 | 6077 |
| C10 | 3278 | 33 | 33 | 3212 |
| C11 | 1068 | 11 | 11 | 1046 |
| C12 | 1927 | 20 | 20 | 1887 |
| C13 | 916 | 10 | 10 | 896 |
| C14 | 1070 | 11 | 11 | 1048 |
| C15 | 7268 | 73 | 73 | 7122 |
| C16 | 1807 | 19 | 19 | 1769 |
| Total | 54,129 | 549 | 549 | 53,031 |

**Table 3.** Category information and corresponding experimental sample size for IN.

| Category | Total | Training | Validation | Test |
|---|---|---|---|---|
| C1 | 46 | 6 | 6 | 30 |
| C2 | 1428 | 172 | 172 | 1084 |
| C3 | 830 | 100 | 100 | 630 |
| C4 | 237 | 29 | 29 | 179 |
| C5 | 483 | 27 | 27 | 429 |
| C6 | 730 | 58 | 58 | 614 |
| C7 | 28 | 4 | 4 | 20 |
| C8 | 478 | 58 | 58 | 362 |
| C9 | 20 | 3 | 3 | 14 |
| C10 | 972 | 117 | 117 | 738 |
| C11 | 2455 | 295 | 295 | 1865 |
| C12 | 593 | 72 | 72 | 449 |
| C13 | 205 | 25 | 25 | 155 |
| C14 | 1265 | 152 | 152 | 961 |
| C15 | 386 | 47 | 47 | 292 |
| C16 | 93 | 12 | 12 | 69 |
| Total | 10,249 | 1238 | 1238 | 7773 |

**Table 4.** Category information and corresponding experimental sample size for ZY.

| Category | Total | Training | Validation | Test |
|---|---|---|---|---|
| C1 | 2625 | 53 | 53 | 2519 |
| C2 | 1302 | 27 | 27 | 1248 |
| C3 | 3442 | 69 | 69 | 3304 |
| C4 | 10,243 | 205 | 205 | 9833 |
| C5 | 1425 | 29 | 29 | 1367 |
| C6 | 1484 | 30 | 30 | 1424 |
| C7 | 1808 | 37 | 37 | 1734 |
| C8 | 1492 | 30 | 30 | 1432 |
| Total | 23,821 | 480 | 480 | 22,861 |

*3.2. Parameter Setting*

In this section, the configuration of the model parameters is introduced at length according to the flowchart of the network of the proposed MOCNN model given in Figure 1. The network parameters settings for 2D octave and 3D DenseNet in the MOCNN model are given in Table 5. For spatial feature extraction, after the input raw 3D cube is down-sampled by PCA, the size of the resulting data cubes is $23 \times 23 \times 9$, $29 \times 29 \times 9$, $21 \times 21 \times 9$, and $27 \times 27 \times 9$ for UP, SV, IN, and ZY, respectively. After that, the reduced data cubes are input to the multi-scale 2D Octave sub-network for spatial feature extraction, where the detailed parameters of the two 2D Octaves in this sub-network configuration are shown in Table 5. For the UP, SV, IN, and ZY, the two convolution kernel sizes $k_{spa}$ in multi-scale 2D octave are set to 5 and 7, 3 and 9, 5 and 7, 3 and 7, respectively. The quantities of convolutional kernel channels $m_{spa}$ are given as 32, 64, 32, and 16 for each 2D octave layer, respectively. For spectral feature extraction, corresponding to the above HSI dataset, the size of the input raw HSI data is $7 \times 7 \times 103$, $7 \times 7 \times 204$, $7 \times 7 \times 220$, and $3 \times 3 \times 80$, respectively. After the channels are weighted by a BAM module, the data are fed into a multi-scale 3D DenseNet to learn the spectrum signatures at various scales. Likewise, the detailed parameters of the two 3D DenseNet in this sub-network configuration are presented in Table 5, where the two convolution kernel size $k_{spe}$ are set to 5 and 7, 5 and 7, 5 and 7, 3 and 5, on UP, SV, IN and ZY, respectively. The number of convolutional kernel channels $m_{spe}$ for each convolutional layer and other detailed configuration settings are also given in Table 5. Besides, the value of the convolution kernel $k_{spe0}$ depends on the value of bands B. When the value of B is even, $k_{spe0}$ takes ceil($B/2 - k_{spe0}$). Conversely, $k_{spe0}$ takes ceil ($B/2 - k_{spe0}$) + 1. The ceil(.) represents the upward rounding calculation function.

**Table 5.** Network parameters settings for 2D octave and 3D DenseNet in the MOCNN model.

| Model | Type / Layer | Filter / Operation | Configuration |
|---|---|---|---|
| 2D octave | 2Doc-conv1 | $(k_{spa}, k_{spa})$, 32 | |
| | 2Doc-conv2 | $(k_{spa}, k_{spa})$, 64 | stride:1, padding:1, BN+mish |
| | 2Doc-conv3 | $(k_{spa}, k_{spa})$, 32 | |
| | 2Doc-conv4 | $(k_{spa}, k_{spa})$, 16 | |
| 3D DenseNet | conv1 | $(1, 1, k_{spe})$, 32 | stride:2, padding:0, BN+mish |
| | conv2 | $(1, 1, k_{spe})$, 16 | stride:1, padding:1, BN+mish |
| | conv3 | $(1, 1, k_{spe})$, 16 | |
| | concat1 | concat(conv2, conv3), 32 | BN+mish |
| | conv4 | $(1, 1, k_{spe})$, 16 | stride:1, padding:1, BN+mish |
| | concat2 | concat(conv2, conv3, conv4), 48 | BN+mish |
| | conv5 | $(1, 1, k_{spe})$, 16 | stride:1, padding:1, BN+mish |
| | concat3 | concat(conv2, conv3, conv4, conv5), 64 | BN+mish |
| | conv6 | $(1, 1, k_{spe0})$, 16 | stride:1, padding:0, BN+mish |

In addition, the parameter settings on four different HSI datasets concerning the learning rate, dropout, batch size, and epochs are shown in Table 6. Due to space considerations,

the parameter settings are not described in detail here, and the optimal parameters derived from the summary of the experiments are given directly.

**Table 6.** Parameter settings on four different datasets inclusive of the learning rate, dropout, batch size, and epoch.

| Class | Learning Rate | Dropout | Batch Size | Epoch |
|-------|---------------|---------|------------|-------|
| UP | 0.00005 | 0.5 | 32 | 250 |
| SV | 0.0001 | 0.5 | 32 | 400 |
| IN | 0.00005 | 0.5 | 32 | 400 |
| ZY | 0.00005 | 0.5 | 64 | 600 |

*3.3. Experimental Setup*

For assessing the validity and superiority of the MOCNN approach, the DL-based classifiers CDCNN [30], FDSSC [31], DBMA [34], DBDA [35], SSAN [24], TriCNN [48], 3DOC-CNN [37], and HRAM [49] are compared with the proposed method.

(1) CDCNN: The CDCNN approach realizes the joint extraction of spectral-spatial features through utilizing multi-scale convolutional filter groups and exploiting the modules of ResNet to significantly improve the depth of the network.

(2) FDSSC: The FDSSC approach utilizes two 3D CNN dense blocks with different kernel sizes, spatial densely block and spectral densely block, to capture deeper spatial and spectral characteristics, respectively.

(3) DBMA: The DBMA approach is a double branch model combined with 3D DenseNet blocks and multiple attention mechanism blocks, applying both attention mechanisms to emphasize important spectral and spatial components.

(4) DBDA: Although DBDA is analogous to the DBMA framework, DBDA employs different size input patches and introduces different channel and spatial attention modules. Moreover, the activation function Mish is introduced to accelerate counter-propagation and prevent overfitting in severely restricted samples.

(5) SSAN: The SSAN approach is a spectral-spatial two-branch two-attention framework model, which adds spectral and spatial attention mechanisms on the basis of Bi-RNN and CNN, respectively.

(6) TriCNN: The TriCNN approach is a three-branch network model based on 3D CNN using various scale convolutional kernels of $1 \times 1 \times 3$, $3 \times 3 \times 1$, and $3 \times 3 \times 3$ sizes to extract spectral, spatial, and spectral-spatial features, respectively, and then fuse the different features obtained from the three sub-networks by feature flattening and concatenation.

(7) 3DOC-CNN: The 3DOC-CNN approach adopts 3Doc-conv to acquire the spectral-spatial characteristics, and introduces attention networks to spotlight the more meaningful characteristics in HSI.

(8) HRAM: The HRAM approach employs a hierarchical residual network to extract spatial and spectral characteristics at the granularity level, utilizing a two-branch structure in parallel with the corresponding convolution kernel. Besides, to boost the discriminative power of the model, it exploits attention mechanisms to assign adaptive weights to spatial and spectral features at various scales.

All experiments were run on a 2080Ti GPU, i9-9900K CPU, 128GB RAM, python 3.6, using tensorflow-gpu 1.14 for implementation. The outcomes reported from all experiments are the average and variance of the results of twenty Monte Carlo runs. To guarantee impartial comparison, the experimental arguments of each classifier are set with default settings of the original paper. In the following, the outcomes of the quantitative comparison experiments between MOCNN and all compared approaches on UP, SV, IN, and ZY are presented in Tables 7–10. Among all classification accuracy comparisons, the highest accuracy is marked in bold. Besides, we visualize the classification maps in Figures 7–10.

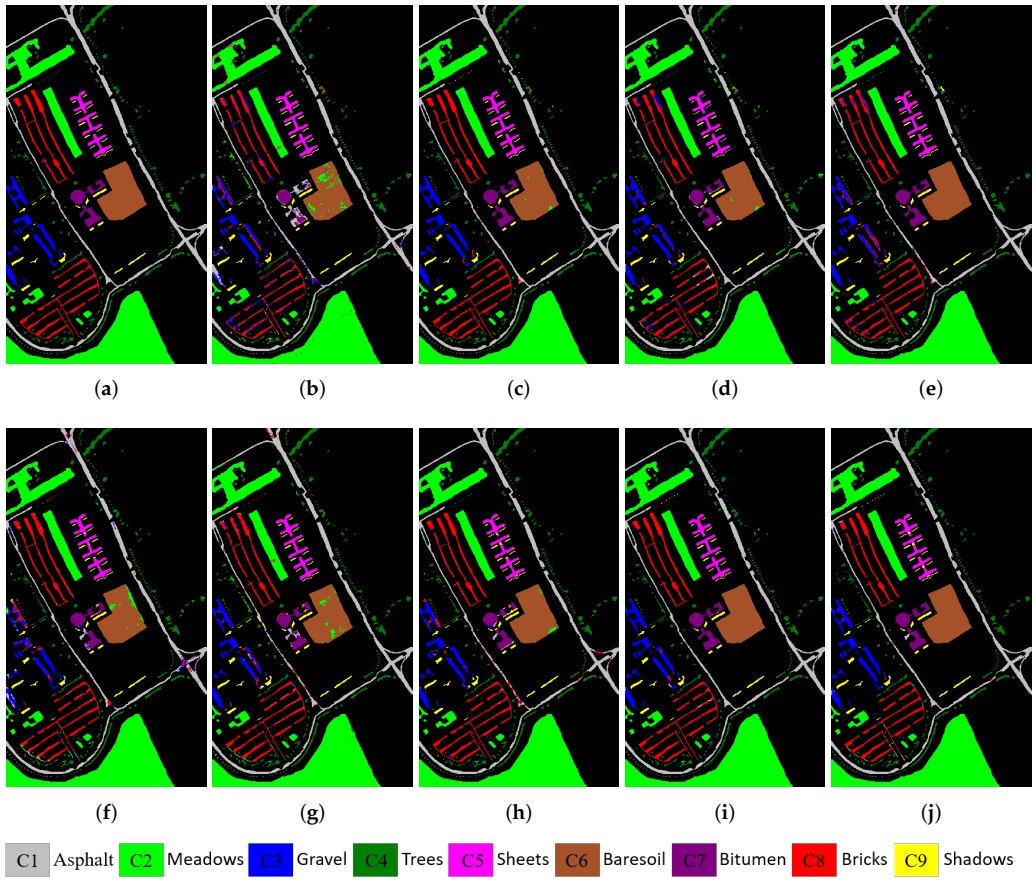

**Figure 7.** Map of the classification results from all approaches on UP. (**a**) Ground truth. (**b**) CD-CNN [30]. (**c**) FDCNN [31]. (**d**) DBMA [34]. (**e**) DBDA [35]. (**f**) SSAN [24]. (**g**) TriCNN [48]. (**h**) 3DOC-CNN [37]. (**i**) HRAM [49]. (**j**) MOCNN.

**Table 7.** Quantitative comparison of the CDCNN [30], FDSSC [31], DBMA [34], DBDA [35], SSAN [24], TriCNN [48], 3DOC-CNN [37], HRAM [49], and MOCNN on UP.

| Category | CDCNN | FDSSC | DBMA | DBDA | SSAN | TriCNN | 3DOC-CNN | HRAM | MOCNN |
|---|---|---|---|---|---|---|---|---|---|
| C1 | 92.19 | **99.35** | 97.67 | 98.90 | 92.80 | 89.86 | 95.20 | 98.75 | 99.21 |
| C2 | 96.37 | 99.70 | 99.30 | 99.72 | 99.45 | 99.71 | **100.00** | 99.68 | 99.95 |
| C3 | 72.28 | 97.03 | 93.10 | 95.97 | 69.73 | 82.95 | 89.19 | 92.87 | **97.25** |
| C4 | 97.94 | 96.61 | 96.62 | 97.53 | **98.15** | 96.71 | 97.91 | 97.36 | 97.98 |
| C5 | 99.19 | **99.70** | 99.38 | 99.51 | 97.01 | 99.87 | 94.55 | 99.29 | 95.32 |
| C6 | 92.04 | 99.66 | 99.07 | 98.55 | 93.66 | 95.15 | 97.99 | 99.89 | **100.00** |
| C7 | 90.47 | **99.95** | 99.14 | 99.27 | 85.43 | 80.53 | 82.56 | 99.26 | 96.20 |
| C8 | 83.18 | 90.79 | 94.30 | 92.63 | 93.17 | 92.93 | **97.62** | 91.85 | 95.52 |
| C9 | 99.08 | 98.15 | 97.03 | 97.59 | 90.73 | 98.79 | 98.80 | 97.72 | **100.00** |
| OA | 92.86 | 98.33 | 98.00 | 98.37 | 94.94 | 95.41 | 97.39 | 98.17 | **98.92** |
|  | ± 2.15 | ± 0.93 | ± 0.39 | ± 0.49 | ± 0.61 | ± 0.86 | ± 0.53 | ± 0.81 | ± **0.55** |
| AA | 91.41 | 97.88 | 97.29 | 97.74 | 91.13 | 92.95 | 94.87 | 97.41 | **97.94** |
|  | ± 3.40 | ± 0.91 | ± 0.53 | ± 0.75 | ± 1.35 | ± 0.92 | ± 0.47 | ± 1.01 | ± **0.71** |
| Ka | 90.48 | 97.79 | 97.35 | 97.85 | 93.27 | 93.91 | 96.54 | 97.57 | **98.57** |
|  | ± 2.91 | ± 1.23 | ± 0.52 | ± 0.65 | ± 0.72 | ± 0.95 | ± 0.65 | ± 1.33 | ± **0.41** |

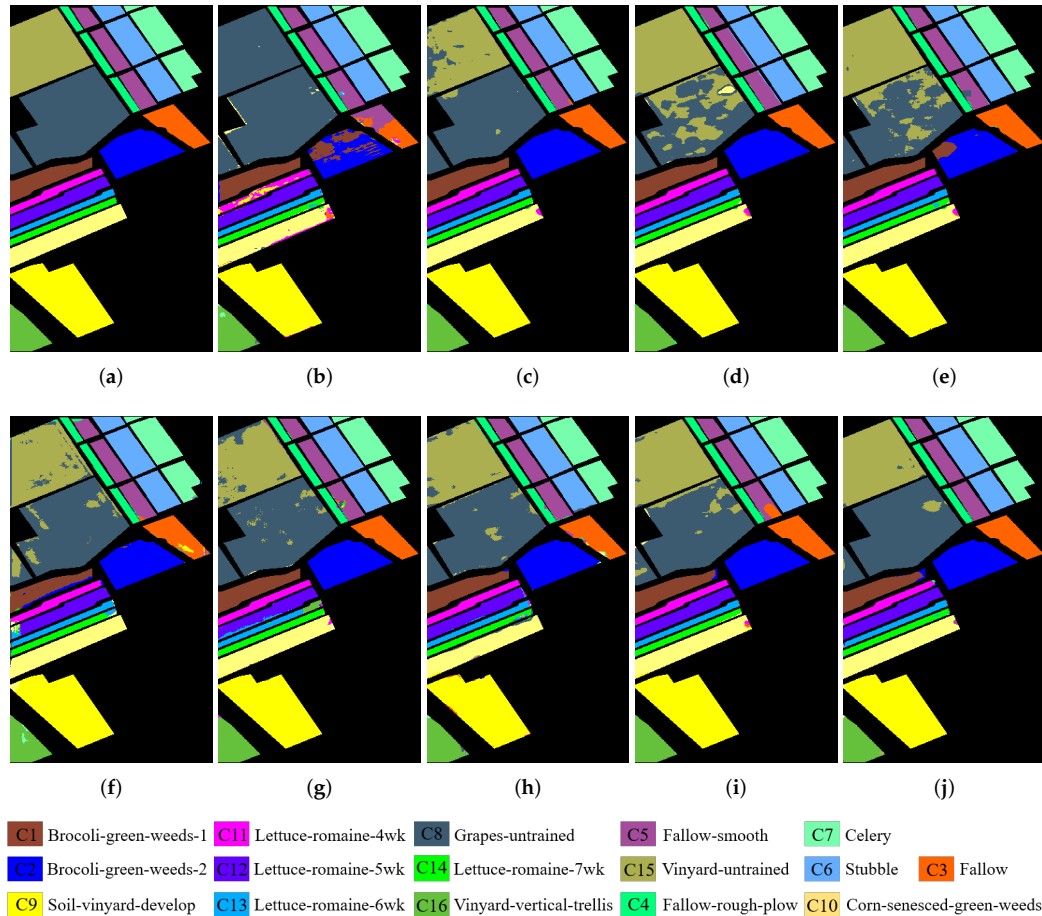

**Figure 8.** Map of the classification results from all approaches on SV. (**a**) Ground truth. (**b**) CD-CNN [30]. (**c**) FDSSC [31]. (**d**) DBMA [34]. (**e**) DBDA [35]. (**f**) SSAN [24]. (**g**) TriCNN [48]. (**h**) 3DOC-CNN [37]. (**i**) HRAM [49]. (**j**) MOCNN.

**Table 8.** Quantitative comparison of the CDCNN [30], FDSSC [31], DBMA [34], DBDA [35], SSAN [24], TriCNN [48], 3DOC-CNN [37], HRAM [49], and MOCNN on SV.

| Category | CDCNN | FDSSC | DBMA | DBDA | SSAN | TriCNN | 3DOC-CNN | HRAM | MOCNN |
|---|---|---|---|---|---|---|---|---|---|
| C1 | 49.99 | **100.00** | **100.00** | 97.48 | 86.32 | 99.98 | 99.08 | **100.00** | **100.00** |
| C2 | 79.97 | **100.00** | 99.92 | 99.96 | 99.53 | 99.92 | 99.78 | 98.72 | **100.00** |
| C3 | 92.96 | 93.64 | 97.76 | 98.20 | 92.51 | 99.49 | 94.16 | 95.99 | **99.70** |
| C4 | 93.95 | 97.55 | 94.23 | 97.13 | **99.78** | 99.50 | 97.66 | 96.78 | 98.66 |
| C5 | 89.59 | 96.73 | 97.51 | 98.46 | 96.84 | 95.58 | 99.28 | 99.47 | **99.78** |
| C6 | 98.18 | 99.81 | 98.77 | 99.88 | **100.00** | **100.00** | 97.68 | 99.99 | 98.70 |
| C7 | 98.02 | 99.99 | 99.80 | 99.91 | 99.03 | 99.96 | 99.60 | 99.93 | **100.00** |
| C8 | 80.45 | 96.25 | 94.66 | 96.99 | 90.88 | 95.81 | 95.70 | 93.49 | **99.83** |
| C9 | 98.20 | 99.72 | **99.77** | 99.40 | 98.47 | 99.88 | 98.68 | 99.44 | 95.78 |
| C10 | 87.17 | 99.40 | 96.47 | 98.31 | 99.66 | 97.61 | 92.28 | 98.60 | **99.67** |
| C11 | 76.08 | 97.00 | 97.34 | 96.88 | 95.32 | 99.91 | 99.71 | 98.16 | **100.00** |
| C12 | 91.28 | 99.88 | 99.71 | 99.70 | 89.51 | 96.23 | 97.14 | **99.98** | 98.28 |
| C13 | 97.01 | **99.87** | 98.15 | 99.37 | 97.43 | 95.87 | 97.43 | 99.47 | 98.62 |
| C14 | 97.35 | 98.13 | 98.40 | 98.46 | 89.60 | 95.85 | 90.36 | 97.64 | **99.89** |
| C15 | 52.71 | 96.11 | 87.93 | 87.12 | 89.96 | 86.58 | 94.02 | 95.26 | **96.47** |
| C16 | 99.02 | **100.00** | 98.13 | 99.96 | 96.10 | 99.12 | 97.06 | 99.93 | 99.35 |

**Table 8.** *Cont.*

| Category | CDCNN | FDSSC | DBMA | DBDA | SSAN | TriCNN | 3DOC-CNN | HRAM | MOCNN |
|---|---|---|---|---|---|---|---|---|---|
| OA | 83.77 ± 5.41 | 97.83 ± 1.10 | 95.91 ± 1.58 | 96.49 ± 2.10 | 94.67 ± 0.53 | 96.59 ± 0.75 | 96.69 ± 0.41 | 97.21 ± 1.12 | **98.69** ± **0.32** |
| AA | 86.31 ± 6.21 | 98.38 ± 0.90 | 97.41 ± 0.95 | 97.95 ± 0.91 | 95.06 ± 0.47 | 97.58 ± 0.71 | 96.85 ± 0.55 | 98.30 ± 0.74 | **99.05** ± **0.55** |
| Ka | 81.95 ± 2.20 | 97.58 ± 1.23 | 95.46 ± 2.05 | 96.10 ± 2.33 | 94.07 ± 0.65 | 96.20 ± 0.84 | 96.31 ± 0.72 | 96.88 ± 1.26 | **98.54** ± **0.71** |

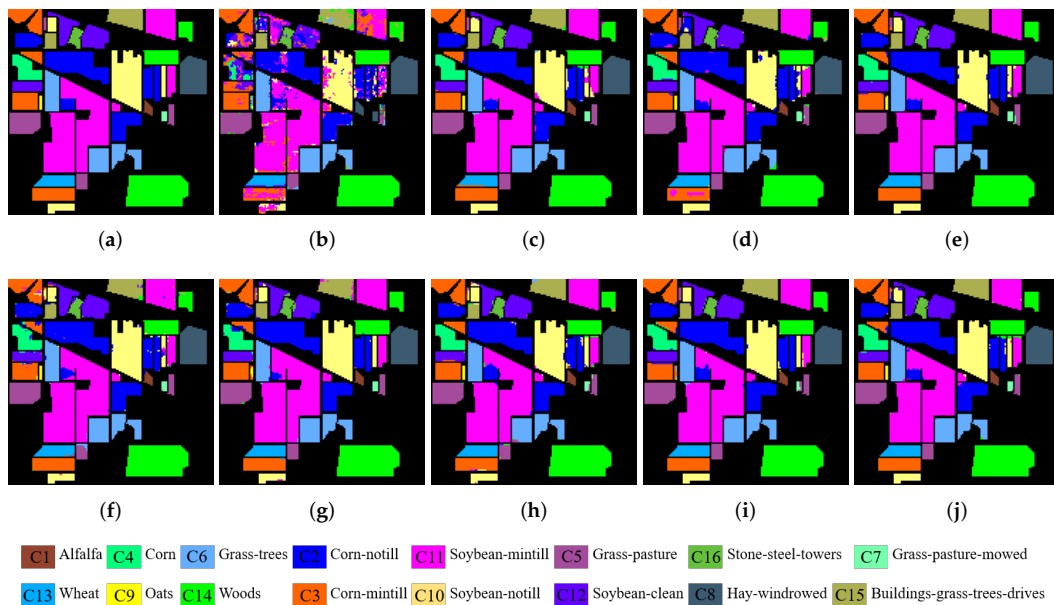

**Figure 9.** Map of the classification results from all approaches on IN. (**a**) Ground truth. (**b**) CD-CNN [30]. (**c**) FDSSC [31]. (**d**) DBMA [34]. (**e**) DBDA [35]. (**f**) SSAN [24]. (**g**) TriCNN [48]. (**h**) 3DOC-CNN [37]. (**i**) HRAM [49]. (**j**) MOCNN.

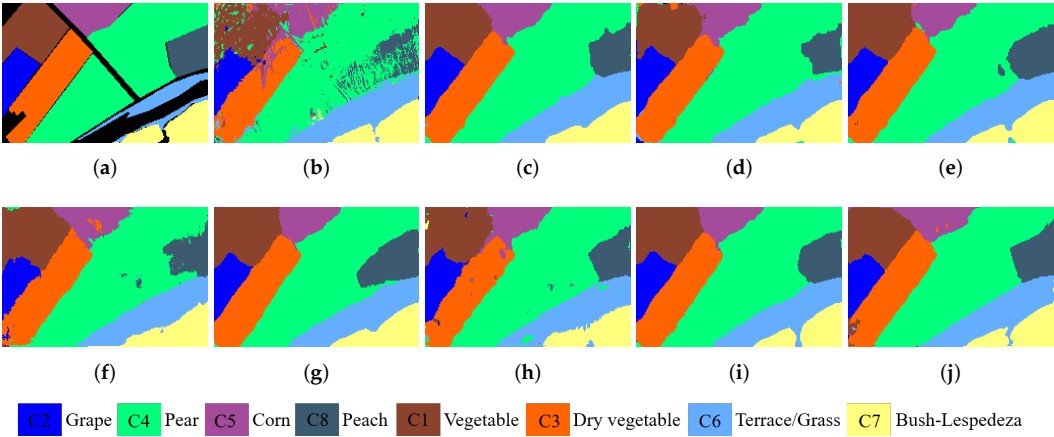

**Figure 10.** Map of the classification results from all approaches on ZY. (**a**) ground truth. (**b**) CD-CNN [30]. (**c**) FDSSC [31]. (**d**) DBMA [34]. (**e**) DBDA [35]. (**f**) SSAN [24]. (**g**) TriCNN [48]. (**h**) 3DOC-CNN [37]. (**i**) HRAM [49]. (**j**) MOCNN.

**Table 9.** Quantitative comparison of the CDCNN [30], FDSSC [31], DBMA [34], DBDA [35], SSAN [24], TriCNN [48], 3DOC-CNN [37], HRAM [49], and MOCNN on IN.

| Category | CDCNN | FDSSC | DBMA | DBDA | SSAN | TriCNN | 3DOC-CNN | HRAM | MOCNN |
|---|---|---|---|---|---|---|---|---|---|
| C1 | 48.27 | 96.07 | 97.35 | **100.00** | 91.18 | **100.00** | **100.00** | 74.24 | 97.06 |
| C2 | 80.85 | **98.26** | 95.51 | 96.64 | 96.76 | 94.59 | 97.48 | 95.76 | 97.42 |
| C3 | 78.45 | 96.97 | **99.13** | 99.10 | 92.72 | 98.35 | 97.99 | 98.24 | 98.57 |
| C4 | 74.12 | 98.45 | 96.23 | 98.46 | 95.63 | 95.45 | **100.00** | 99.04 | **100.00** |
| C5 | 96.44 | 97.99 | 97.64 | **98.77** | 92.53 | 97.81 | 94.13 | 97.03 | 95.10 |
| C6 | 96.27 | 97.41 | 98.69 | 98.92 | **99.30** | 99.69 | 97.36 | 98.36 | 98.74 |
| C7 | 44.91 | 95.89 | 81.70 | 82.30 | **100.00** | 99.20 | 75.00 | 70.70 | 95.00 |
| C8 | 89.30 | **100.00** | 99.82 | 99.95 | **100.00** | **100.00** | **100.00** | 98.49 | **100.00** |
| C9 | 42.50 | 95.63 | 95.82 | 97.20 | **100.00** | 63.33 | **100.00** | 60.00 | **100.00** |
| C10 | 78.95 | 96.16 | 95.92 | 94.41 | 96.31 | 97.66 | **98.15** | 96.92 | 97.97 |
| C11 | 78.02 | 99.03 | 97.10 | 99.18 | **99.16** | 98.27 | 97.86 | 99.15 | 98.98 |
| C12 | 63.40 | 98.68 | 94.92 | 96.49 | 96.31 | 92.27 | 94.79 | 95.18 | 98.22 |
| C13 | 98.45 | 98.69 | 99.79 | 98.78 | 94.97 | 99.56 | 99.37 | **100.00** | **100.00** |
| C14 | 93.75 | 99.12 | 98.69 | 99.32 | 99.29 | 98.99 | 99.19 | 99.30 | **99.90** |
| C15 | 86.86 | 98.13 | 95.94 | 97.90 | 94.33 | 94.99 | 94.67 | 98.69 | **100.00** |
| C16 | 97.64 | **95.98** | 93.18 | 91.66 | 87.32 | 95.12 | 94.37 | 94.00 | 95.65 |
| OA | 81.76 ± 3.71 | 98.16 ± 0.80 | 97.04 ± 1.08 | 97.88 ± 0.54 | 97.14 ± 0.52 | 97.36 ± 0.49 | 97.60 ± 0.32 | 97.79 ± 1.50 | **98.58 ± 0.41** |
| AA | 78.01 ± 5.81 | 97.65 ± 1.04 | 96.09 ± 0.84 | 96.82 ± 0.49 | 95.99 ± 1.12 | 95.33 ± 1.53 | 96.27 ± 0.91 | 92.19 ± 3.35 | **98.29 ± 0.67** |
| Ka | 79.08 ± 4.36 | 97.91 ± 1.01 | 96.62 ± 1.24 | 97.58 ± 1.07 | 96.74 ± 0.81 | 96.99 ± 0.55 | 97.27 ± 0.55 | 97.48 ± 1.01 | **98.39 ± 0.51** |

**Table 10.** Quantitative comparison of the CDCNN [30], FDSSC [31], DBMA [34], DBDA [35], SSAN [24], TriCNN [48], 3DOC-CNN [37], HRAM [49], and MOCNN on ZY.

| Category | CDCNN | FDSSC | DBMA | DBDA | SSAN | TriCNN | 3DOC-CNN | HRAM | MOCNN |
|---|---|---|---|---|---|---|---|---|---|
| 1 | 80.19 | 97.34 | 96.86 | 96.88 | **99.88** | 95.28 | 96.10 | 95.67 | 97.66 |
| 2 | 85.53 | 97.63 | 96.64 | 97.03 | 90.46 | 97.75 | **99.28** | 97.58 | 97.04 |
| 3 | 81.97 | 98.10 | 97.06 | 98.09 | 97.31 | 95.52 | 92.05 | 98.01 | **99.49** |
| 4 | 89.22 | 98.38 | 98.52 | 98.46 | 96.68 | 97.66 | **98.66** | 98.47 | 97.67 |
| 5 | 55.62 | 95.81 | 92.46 | 96.10 | 94.15 | 90.49 | 91.21 | 97.41 | **99.93** |
| 6 | 88.00 | 96.37 | 95.75 | 95.13 | 94.03 | 96.27 | 91.20 | 97.09 | **98.03** |
| 7 | 94.75 | 96.27 | 98.87 | 98.49 | **99.60** | 97.36 | 98.44 | 97.62 | 97.98 |
| 8 | 61.69 | 94.29 | 94.31 | 94.54 | 86.59 | 92.19 | 95.10 | 95.33 | **98.95** |
| OA | 84.14 ± 3.73 | 97.43 ± 0.83 | 96.96 ± 0.44 | 97.50 ± 0.55 | 96.06 ± 0.95 | 95.98 ± 0.66 | 96.31 ± 0.45 | 97.59 ± 0.36 | **98.16 ± 0.63** |
| AA | 86.31 ± 6.21 | 96.77 ± 0.62 | 95.93 ± 0.68 | 96.84 ± 0.73 | 94.84 ± 1.02 | 95.31 ± 0.81 | 95.25 ± 0.41 | 97.14 ± 0.41 | **98.34 ± 0.62** |
| Ka | 81.95 ± 2.20 | 96.63 ± 1.12 | 96.02 ± 0.57 | 96.72 ± 0.72 | 94.83 ± 1.80 | 94.74 ± 0.85 | 95.15 ± 0.77 | 96.84 ± 0.47 | **97.59 ± 1.04** |

*3.4. Classification Maps and Results*

The accuracy metrics, including OA, AA, and Ka, for the UP dataset from our proposed MOCNN approach and several other DL-based comparison models are provided in Table 7, while corresponding classification maps are given in Figure 7.

From Table 7, it can be seen that the MOCNN approach achieves the optimal classification precision, with values of 98.92% for OA, 97.94% for AA, and 98.57% for Ka. The CDCNN approach based on multi-scale convolutional filters and ResNet structure results in the worst accuracy, with 92.86% OA, 91.41% AA, and 90.48% Ka. The reason may be that the limitation of this network structure design leads to a weak feature extraction ability of

the model under the limited training samples. Owing to the fact that the Bi-RNN model based on the attention mechanism effectively enhances the extraction of spectral features, the OA of the SSAN approach is 2.08% higher than the CDCNN. However, since SSAN employs the traditional CNN network for spatial feature extraction, it lacks the adequate exploitation of spatial features from convolutional layers of various depths. The TriCNN increases the accuracy of OA by 2.55% and 0.47% over CDCNN and SSAN, respectively, which may be owing to three convolutional kernels of various scales utilized by TriCNN and its strengthened ability of the network to explore complex features. Nevertheless, TriCNN still also lacks the utilization of characteristics across various convolutional layers. The HRAM effectively improves the utilization of hierarchical features by using hierarchical ResNets, and its OA accuracy enhances by 2.76% compared with TriCNN. Similarly, the FDSSCN, DBMA, and DBDA approaches employ 3D DenseNet to improve the reuse and propagation of hierarchical characteristic information and further enhance the capability of the network for feature extraction. Compared with SSAN, the OA accuracy of FDSSCN, DBMA, and DBDA approaches improved by 3.39%, 3.06%, and 3.43%, respectively. Comparing our method with 3DOC-CNN using single-scale octave convolution alone, OA improves by 1.53% and the accuracy of the two metrics (AA and Ka) is also superior to that of 3DOC-CNN. In spite of the fact that the proposed MOCNN method achieves worse results than 3DOC-CNN in certain categories, such as Meadows and Bricks, while the classification accuracy is the highest in other categories.

It is evident from the corresponding Figure 7 that the classification maps of the MOCNN and DBDA approaches are significantly superior to other methods, with better homogeneity and smoother classification maps for the "Baresoil" category than those generated by the other methods. Additionally, our proposed approach contains fewer misclassified pixels, and noticeably better classification maps in the "Gravel" category. That is because the MOCNN combining multi-scale 2D octave convolution and multi-scale 3D DenseNet not only takes into account the utilization of features across hierarchical convolution layers, but also sufficiently exploits the complex characteristics at various scales.

As shown in Table 8, the MOCNN approach realizes the best categorization results with 98.69% OA, 99.05% AA, and 98.54% Ka. It is notable that CDCNN has the worst classification accuracy, which is 14.92% lower than our method. This also indicates the fact that limitations of CDCNN may exist in the design of the network structure, failing to exploit the superiority of multi-scale convolutional filters and residual network structures to mine distinguishing characteristics at various scales from complicated scenes. Although TriCNN and SSAN methods are both traditional CNN-based in spatial feature extraction, the OA accuracy of TriCNN is 96.59%, which is 1.92% higher than the value of 94.67% gained by SSAN. In the corresponding categorization maps in Figure 8, TriCNN has significantly fewer misclassification noise points in terms of spatial details than SSAN. The reason for this is that TriCNN adopts a three-branch network with convolutional kernels of various scales, which greatly boosts the network's ability to capture complex characteristics. The HRAM method based on hierarchical ResNet achieves an OA accuracy of 97.21%, which exceeds the conventional CNN-based TriCNN and SSAN, respectively, by 0.62% and 2.54%. In addition, in comparison to FDSSC, DBMA, and DBDA based on single-scale DenseNet, our method's classification accuracy OA is 0.86%, 2.78%, and 2.20% higher, respectively. This may be explained by the fact that MOCNN uses not only DenseNet structures but also multi-scale convolutional kernels, which further facilitate the extraction of complex characteristic information at various scales. Notably, despite the fact that MOCNN achieves optimal categorization accuracy in many categories, certain categories in other comparison methods, such as FDSSC in categories "Brocoli-green-weeds-1", "Brocoli-green-weeds-2", and "Vinyard-vertical-trellis", also obtain 100% categorization accuracy. It demonstrates that other DL-based models also have strong feature learning capabilities and may assign more weights to certain categories to realize accurate classification of that category.

Meanwhile, as seen in the corresponding classification map in Figure 8, the MOCNN approach provides the most accurate categorization result map and is smoother in homoge-

neous regions. Its causes can be attributed to the following: first, the densely connected based structure merges features from convolutional layers of various depths, enhancing the reuse and propagation of feature information. Then, the use of multi-scale convolutional kernels facilitates the extraction of complex characteristic information at various scales. In addition, two attention mechanism modules are utilized to highlight important spatial feature information and reinforce the extraction of feature channel information in the network, respectively.

As shown in Table 9, MOCNN achieves the optimal categorization performance in OA, AA, and Ka, achieving 98.58%, 98.29%, and 98.39%. Compared with the suboptimal classification results, our method improves over the FDSSC by 0.42%, 0.64%, and 0.48%. Compared with HRAM and TriCNN, the accuracy improvement of AA by MOCNN is more remarkable, i.e., 6.1% and 2.96% higher, respectively. This is primarily attributed to employing a WCL-based sample balancing strategy that allows MOCNN to allocate appropriate weights to each category and to pay more attention to the categories with small sample sizes. Furthermore, it is clear from Table 3 that the sample distribution of the IN is highly unbalanced. Certain categories, such as "Soybean-mintill" and "Corn-notill", have 295 and 172 samples. Yet, there are also some categories with only a few or dozens, such as "Oats", "Grass-pasture-mowed", "Alfalfa", and "Corn", which have only 3,4, 6, and 29 samples, respectively. Nonetheless, the MOCNN also achieved an OA accuracy of 100.00%, 95.00%, 97.06%, and 100.00% for these categories, respectively. It illustrates that the compound loss function based on WCL proposed in this paper, allocating appropriate loss weights to various categories, can make the model more focused on the categories with small samples, which effectively alleviates the problem of poor classification results due to sample imbalance. As we can see from Figure 9, although the MOCNN method has few misclassified pixels on the categories Corn-notill, Corn-mintill, and Soybean-mintill, the overall classification results are relatively less noisy than the other approaches. Moreover, the smoothing in homogeneous areas (such as Corn-notill and Grass-pasture-mowed) is significantly better than the other methods.

As seen in Table 10, the MOCNN approach still realizes the best categorization results with 98.16% OA, 98.34% AA, and 97.59% Ka. Although the categorization accuracy of MOCNN is not the highest in the "Grape", "Peach", "Corn", and "Terrace/Grass regions", the accuracy of our method exceeds 97.04% in each of these categories. This illustrates well the effectiveness of our method in extracting discriminable features between different categories. Moreover, we can see from Figure 10 that the classification map of the CDCNN approach with the lowest categorization accuracy has a large number of misclassified pixels and pepper noise. The OA accuracy of SSAN and 3DOC are 96.06% and 96.31%, respectively, which are both better than 84.14% of CDCNN. In contrast, SSAN and 3DOC-CNN also have fewer mislabels, but the classification results in the "Pear" and "Dry vegetable" regions are still not satisfactory. Conversely, in the densely connected based approaches, i.e., FDSSC, DBMA, DBDA and MOCNN, categorization results are significantly better than the previous two methods. This illustrates that the complex spatial structure information of HSI can be effectively exploited by using the complementary yet related information between the features of different convolutional layers. Besides, the ResNet-based HRAM method achieves 97.59% categorization accuracy with less classification noise on the categorization map. Nonetheless, since it adopts the single-scale convolutional kernel, it suffers from limitations in capturing boundary detail information. However, the MOCNN method not only produces a smoother appearance in homogeneous areas but also has clear boundaries and preserves edge detail information well, which also demonstrates that the proposed MOCNN based on multi-scale convolutional structure and attention network mechanism can adequately capture complex characteristic information at various scales.

*3.5. Discussion*

In this section, we first discuss the classification performance of the proposed method and other research methods with varying training sample sizes. Following that, we discuss

the effects of different functional modules on the classification performance of the model in the ablation experiments.

### 3.5.1. Performance under Different Numbers of Training Samples

DL is a data-driven approach that relies on the availability of massive amounts of labeled data. Therefore, to further investigate the categorization performance under various numbers of training samples, we conducted extensive experiments on UP, SV, IN, and ZY using different percentages of training samples, respectively. Figure 11 exhibits the corresponding experimental results. Notably, since the highest values of CDCNN on the IN and ZY are lower than the minimum coordinate values on the corresponding sub-figures, the plot of CDCNN is not shown in the corresponding sub-figures. As anticipated, the categorization precision of the DL-based approach improves along with the increase in the number of training samples. As seen in Figure 11, it is evident that our method can still achieve superior classification precision with limited training samples. Hence, our method can save a great deal of labor and cost in labeling samples.

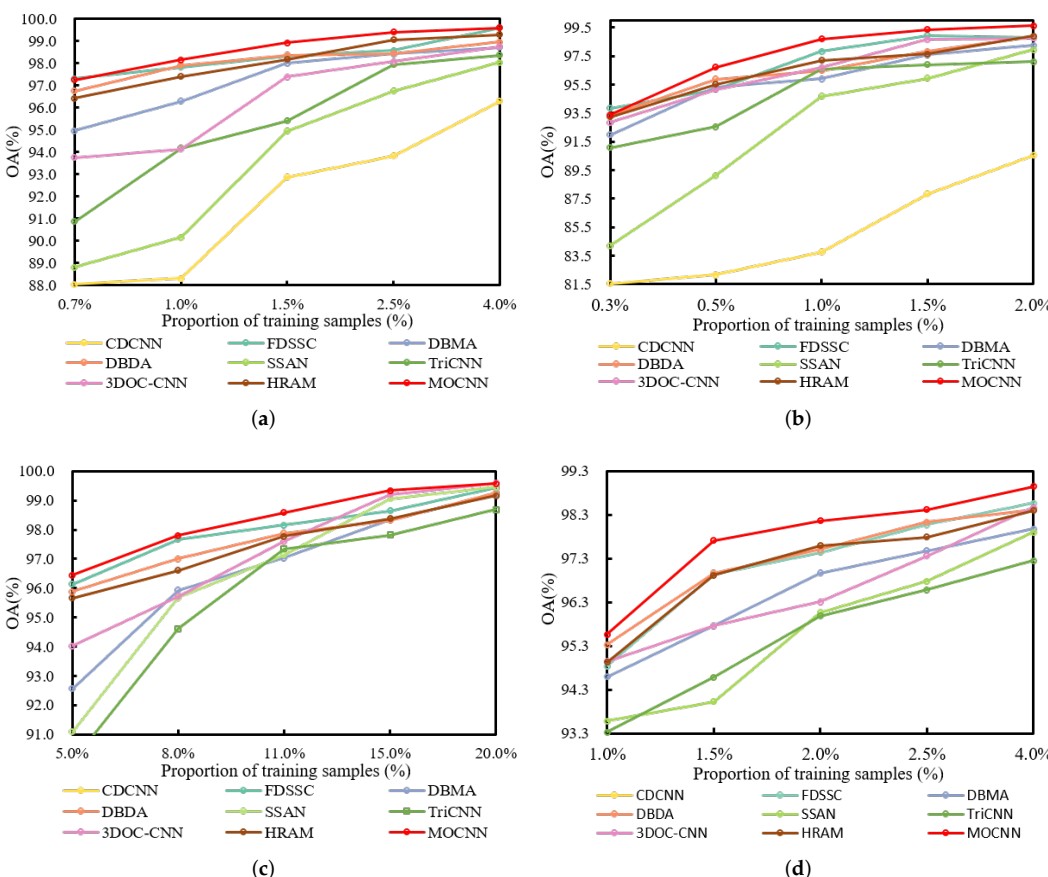

**Figure 11.** Classification performance under different percentages of training samples. (**a**) UP. (**b**) SV. (**c**) IN. (**d**) ZY.

### 3.5.2. Ablation Experiment

In this section, ablation experiments were designed to validate the effectiveness of the spectral attention module BAM, the channel attention mechanism ECA, multi-scale 3D DenseNet, and multi-scale 2D octave. Specifically, no-ECA denotes that the proposed model does not use the ECA module, and only the BAM module is employed. Similarly, no-BAM denotes that the proposed method utilizes only the ECA module. MONA indicates that ECA and BAM attention networks are not used in the proposed MOCNN approach. The MCNN denotes that the proposed MOCNN method only adopts multi-scale 3D DenseNet for spectral feature extraction, and there is no branch network employed for

spatial information extraction. The MONN denotes that the proposed MOCNN method only adopts multi-scale 2D octave for feature extraction, and lacks the network for the extraction of spectral information.

From Figure 12, the classification accuracy of the MONA approach without using any attention mechanism is significantly lower than that of the no-ECA and no-BAM approaches on the four datasets, thus fully proving the effectiveness of the ECA and BAM attention modules. In addition, it is apparent that the classification outcomes of MONN and MCNN are significantly lower than MOCNN, which is due to the fact that MONN based on multi-scale 2D octave can only extract spatial feature information at various scales, while lacking access to spectral information. Similarly, MCNN based on multi-scale 3D DenseNet can only withdraw spectral signatures at various scales and lacks the mining of complex spatial information. It also further demonstrates that the single extraction network, either for spectral features alone or for spatial feature information alone, fails to simultaneously sufficiently exploit the spectral-spatial feature information that is favorable for HSI categorization.

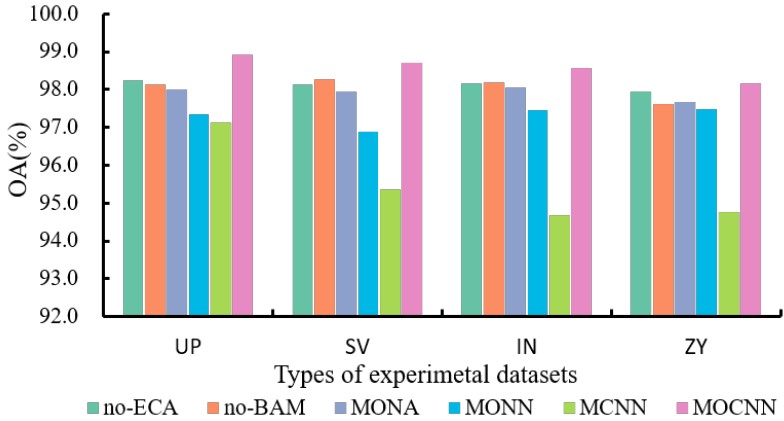

**Figure 12.** Results of ablation experiments on different types of datasets.

## 4. Conclusions

In this study, a new multi-scale spectral-spatial attention network combining 2D octave and 3D CNN is proposed for HSI classification. Concerning spatial features, based on 2D octave, multi-scale 2D octave is proposed to obtain complicated spatial characteristic information. It can not only adequately mine the spatial feature information under the complex structure by using convolutional kernels of various scales, but also decompose the obtained feature map into LF and HF components to lessen the redundancy of spatial feature information. Concerning spectral features, we use the multi-scale 3D DenseNet to sufficiently extract discriminative spectral characteristics at various scales, while fusing spectral features in both shallow and deeper convolutional layers to enhance the transmission and reuse of feature information among various convolutional layers. Besides, Two attention models (BAM and ECA) are used to improve network performance. Among them, BAM is exploited to assign proper weight values for each spectral band while suppressing insignificant spectral bands to alleviate the effect of redundant HSI bands in the classification. ECA is used in the two feature extraction sub-networks to enhance the interactions of feature information among feature channels and boost the feature extraction capability. Moreover, a sample balancing strategy based on WCL is applied to address the problem of sample imbalance. Experimental outcomes indicate that the proposed MOCNN approach outperforms several other compared approaches for classification.

In the future, we will develop semi-supervised or unsupervised HSI classification methods able to operate in scenarios with limited and imbalanced samples.

**Author Contributions:** L.L. and S.Z. proposed the core framework design idea of MOCNN network. L.L., S.Z. and Z.C. participated in conducting the code implementation, experimental analysis, and manuscript writing. J.L. and A.P. provided valuable advice on the methodology and carefully modified the manuscript. All authors have read and agreed to the published version of the manuscript.

**Funding:** This research is supported by the National Natural Science Fund of China (T2225019), the Jiangxi Province Key Subject Academic and Technical Leader Funding Project (20225BCJ23019), the Jiangxi Provincial Natural Science Foundation (20224BAB202007 and 20224ACB202002), the Key R & D Program of Hunan Province (2019SK2102), and the China Scholarship Council.

**Conflicts of Interest:** The authors declare no conflict of interest.

## Abbreviations

The following abbreviations are used in this manuscript:

| | |
|---|---|
| CNN | convolutional neural network |
| HSI | hyperspectral image |
| MLR | multi-nomial logistic regression |
| MRF | Markov random field |
| DL | deep learning |
| 3Doc-conv | 3D octave convolution |
| MOCNN | multi-scale spectral-spatial attention network framework combining 2D octave and 3D CNNs |
| multi-scale 3D DenseNet | multi-scale DenseNet based on 3D CNNs |
| multi-scale 2D octave | multi-scale 2D octave convolution network |
| LF | low frequency |
| HF | high frequency |
| PCA | principal component analysis |
| BAM | band attention mechanism |
| ECA | efficient channel attention mechanism |
| WCL | weighted cross-entropy loss function |
| AP | averaging pooling |
| BN | batch normalization |
| 2Doc-conv | 2D octave convolution |
| OA | overall accuracy |
| AA | average accuracy |
| Ka | Kappa cofficient |
| UP | Pavia University dataset |
| SV | Salinas Valley dataset |
| IN | Indian Pines dataset |
| ZY | Zaoyuan dataset |
| SR | spatial resolution |

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
