# Peer review of "Multi-Scale Spectral-Spatial Attention Network for Hyperspectral Image Classification Combining 2D Octave and 3D Convolutional Neural Networks"

_remotesensing, doi:10.3390/rs15071758_

Round 1
Reviewer 1 Report
I have the following general comments and suggestions.
The structure of paper should be improved.
You should add a Discussion section where you should compare you results with other studies in the field.
The introduction can be written in more clearer way. You should add a separate table with all abbreviations used in the text for easier reading.
General advice for the Figures is that they are too small and difficult to read you should increase the font of Figures, especially Fig. 1,2,3,4.
I advice you to add slightly more description of the other methods used in Table 8 and to analyze more deeply the results presented in Table 8, it will be interesting why in some cases the other methods achieve 100.00
In the paper there are lots of punctuation errors, I recommend extensive Editing of English and punctuation.
More precisely you should add a comma after which of the formulas-
on page 4 '','' is necessary after formula (1) and (2) before where
on page 5 '','' should be add after formula (3), (4) and (7);
and ''.'' after formula (5) and fomula (6).
on page 6 '','' should be add after (8)
on page 7 '','' should be add after (9),(10) and (11)
on page 8 '','' should be add after (12),(13),(14) and (15)
on page 9 '','' should be add after (16) and (17)
On many places missing space is necessary for example:
on page 4 line 28 (AP)operation
on page 5 line 27 problem[25] and ResNet[24]
on page 12 line 19 takes ceil(B/2-kspe0)
on page 13 line 2 CDCNN[26]
on page 13 the description of Table 7the second line should not start with the comma.
It is not good style to start the sentences with [], see for example page 2 line 37 [28] proposed should be In [28] it is proposed. The same on line 39 and on page 3 line 5 and line 7.
When you talk about Markov random fields on page 2 line 12 I suggest to add some more sentences about their significance and I recommend to cite the following papers regarding them:
Dimov, I., et al. An unbiased Monte Carlo method to solve linear Volterra equations of the second kind. Neural Comput & Applic 34, 1527–1540 (2022). https://doi.org/10.1007/s00521-021-06417-5
T. Koyakumaru, M. Yukawa, E. Pavez and A. Ortega, "A Graph Learning Algorithm Based On Gaussian Markov Random Fields And Minimax Concave Penalty," ICASSP 2021 - 2021 IEEE International Conference on Acoustics, Speech and Signal Processing (ICASSP), Toronto, ON, Canada, 2021, pp. 5410-5414, doi: 10.1109/ICASSP39728.2021.9413850.
Todorov, V., Dimov, I., Ostromsky, T., Zlatev, Z., Georgieva, R., Poryazov, S. (2022). Optimized Quasi-Monte Carlo Methods Based on Van der Corput Sequence for Sensitivity Analysis in Air Pollution Modelling. In: Fidanova, S. (eds) Recent Advances in Computational Optimization. WCO 2020. Studies in Computational Intelligence, vol 986. Springer, Cham. https://doi.org/10.1007/978-3-030-82397-9_20
Tariq, A.; Shu, H. CA-Markov Chain Analysis of Seasonal Land Surface Temperature and Land Use Land Cover Change Using Optical Multi-Temporal Satellite Data of Faisalabad, Pakistan. Remote Sens. 2020, 12, 3402. https://doi.org/10.3390/rs12203402
The Conclusion section is too short, you should stress on the novelties of your approach.
The major contributions from the Introduction on page 3 should be add here.
Author Response
We gratefully acknowledge the Reviewers for their detailed and constructive criticisms, which have greatly helped us to improve the quality and presentation of our manuscript. In the following, we provide detailed, item-by-item, point-by-point responses to all the issues raised by the Anonymous Reviewers.
Responses to comments by Reviewer
I have the following general comments and suggestions.
1.The structure of paper should be improved. You should add a Discussion section where you should compare you results with other studies in the field.
Thanks for the comment. We are sorry for not expressing this clearly in the manuscript. Following the Reviewer’s outstanding suggestion, it has been done in the revised manuscript as suggested. We have added a discussion of its comparison with other research methods while comparing the experimental results. Corresponding content has been marked in red. (see page 15, lines 372-396; see pages 16-17, lines 398-441; see page 19, lines 481-465) In addition, we have added a “Discussion” section in the “Experimental Results and Discussion” section. In this section, we first discuss the classification performance of the proposed method and other research methods with varying training sample sizes. Following that, we discuss the effects of different functional modules on the classification performance of the model in the ablation experiments. Corresponding content has been marked in red. (see pages 20-22, lines 465-500) Thanks again! This is very helpful to improve the quality of my paper.
2.The introduction can be written in more clearer way. You should add a separate table with all abbreviations used in the text for easier reading.
This is a very good suggestion indeed. In the revised manuscript, we have revised manuscript as suggested by adding an “Abbreviations” section. Corresponding content has been marked in red. (see pages 22-23, lines 527-530) Thank you again for your outstanding advice, this suggestion has greatly helped us to improve the technical quality and presentation of the manuscript.
3.General advice for the Figures is that they are too small and difficult to read you should increase the font of Figures, especially Fig. 1,2,3,4.
We apologize for these issues and gratefully thank the Reviewer for his/her careful reading of our manuscript. It has been done in the revised manuscript as suggested. At the same time, we have also carefully checked the entire paper again for similar problems. (see pages 5-8, Fig. 1,2,3,4 and 5)
4.I advice you to add slightly more description of the other methods used in Table 8 and to analyze more deeply the results presented in Table 8, it will be interesting why in some cases the other methods achieve 100.00
This is a very good suggestion. Following the Reviewer’s outstanding suggestion, we have added more descriptions of the other methods and analyzed more deeply the results in Table 8. (see pages 16-17, lines 397-425)
5.In the paper there are lots of punctuation errors, I recommend extensive Editing of English and punctuation.
More precisely you should add a comma after which of the formulas-on page 4 '','' is necessary after formula (1) and (2) before where
on page 5 '','' should be add after formula (3), (4) and (7);
and ''.'' after formula (5) and fomula (6).
on page 6 '','' should be add after (8)
on page 7 '','' should be add after (9),(10) and (11)
on page 8 '','' should be add after (12),(13),(14) and (15)
on page 9 '','' should be add after (16) and (17)
On many places missing space is necessary for example:
on page 4 line 28 (AP)operation
on page 5 line 27 problem[25] and ResNet[24]
on page 12 line 19 takes ceil(B/2-kspe0)
on page 13 line 2 CDCNN[26]
on page 13 the description of Table 7the second line should not start with the comma.
It is not good style to start the sentences with [], see for example page 2 line 37 [28] proposed should be In [28] it is proposed. The same on line 39 and on page 3 line 5 and line 7.
We apologize for these mistakes and bad descriptions, while we gratefully thank the Reviewer for his/her careful reading of our manuscript. It has been done in the revised manuscript as suggested. We have gone through the whole manuscript and modified some mistakes. Furthermore, in the “Introduction” section, the corresponding content has been marked in red. (see pages 2-3, lines 17-30, 45-83)
6.When you talk about Markov random fields on page 2 line 12 I suggest to add some more sentences about their significance and I recommend to cite the following papers regarding them:
Dimov, I., et al. An unbiased Monte Carlo method to solve linear Volterra equations of the second kind. Neural Comput & Applic 34, 1527–1540 (2022). https://doi.org/10.1007/s00521-021-06417-5
T. Koyakumaru, M. Yukawa, E. Pavez and A. Ortega, "A Graph Learning Algorithm Based On Gaussian Markov Random Fields And Minimax Concave Penalty," ICASSP 2021 - 2021 IEEE International Conference on Acoustics, Speech and Signal Processing (ICASSP), Toronto, ON, Canada, 2021, pp. 5410-5414, doi: 10.1109/ICASSP39728.2021.9413850.
Todorov, V., Dimov, I., Ostromsky, T., Zlatev, Z., Georgieva, R., Poryazov, S. (2022). Optimized Quasi-Monte Carlo Methods Based on Van der Corput Sequence for Sensitivity Analysis in Air Pollution Modelling. In: Fidanova, S. (eds) Recent Advances in Computational Optimization. WCO 2020. Studies in Computational Intelligence, vol 986. Springer, Cham. https://doi.org/10.1007/978-3-030-82397-9_20
Tariq, A.; Shu, H. CA-Markov Chain Analysis of Seasonal Land Surface Temperature and Land Use Land Cover Change Using Optical Multi-Temporal Satellite Data of Faisalabad, Pakistan. Remote Sens. 2020, 12, 3402. https://doi.org/10.3390/rs12203402
This is a very good comment indeed. It has been done in the revised manuscript as suggested. When we talked about the Markov random field, we have added some descriptions about its significance. Corresponding content has been marked in red. (see page 2, lines 19-25) In addition, we also have cited relevant papers about them as suggested.
7.The Conclusion section is too short, you should stress on the novelties of your approach. The major contributions from the Introduction on page 3 should be add here.
Thanks for the comment. It has been done in the revised manuscript as suggested. We have re-capitulated the major contributions and novelties of our paper. Corresponding content has been marked in red. (see page 22, lines 502-519)
Last but not least, we would like to take this opportunity to gratefully thank the reviewer again for his/her outstanding comments and suggestions, which greatly helped us to improve the technical quality and presentation of our manuscript.

Reviewer 2 Report
This paper proposes a hyperspectral image classification method using 2D Octave and 3D CNNs. Overall, the proposed model is clearly presented and the experiments show the effectiveness of it. Some minor suggestions are as follows.
1. What is the difference between channel and spectral attention mechanisms? It seems that both of them share the spirit of weighting the input channels.
2. In the experiments, it would be better to compare with more related models proposed in recent two years.
3. In the ablation study, it would be interesting to see the performance with only multi-scale 3D DenseNet or only multi-scale 2D Octave.
4. Some very related works may be considered to cite, such as 'Hyperspectral Image Classification With Attention-Aided CNNs, 2021 TGRS', 'Cross-Modality Contrastive Learning for Hyperspectral Image Classification, 2022 TGRS, 10.1109/TGRS.2022.3188529', 'Multiscale Progressive Segmentation Network for High-Resolution Remote Sensing Imagery, 2022 TGRS, DOI 10.1109/TGRS.2022.3207551'.
Author Response
Responses to comments by Reviewer
This paper proposes a hyperspectral image classification method using 2D Octave and 3D CNNs. Overall, the proposed model is clearly presented and the experiments show the effectiveness of it. Some minor suggestions are as follows.
We would like to take this opportunity to gratefully thank the reviewer for his/her careful assessment and summary of the main contributions of our work. We also gratefully thank the reviewer for taking the time to review our contribution and providing very valuable suggestions.
1.What is the difference between channel and spectral attention mechanisms? It seems that both of them share the spirit of weighting the input channels.
Thanks for the comment. We are sorry for not expressing this clearly in the manuscript. Following the Reviewer’s outstanding suggestion, we have added the detailed description in the paper. The spectral attention mechanism module is used to allocate proper weight values for each spectral band while suppressing insignificant spectral bands to mitigate the effect of redundant HSI bands in the classification. The channel attention mechanism module is used in the two feature extraction sub-networks to enhance the interactions of feature map information among feature channels (instead of spectral bands) and improve the feature extraction capability.
Corresponding revised content has been marked in red. Thanks again! This is very helpful to improve the quality of my paper. (see page 3, lines 95-99; see pages 7-8, lines 213-224, lines 234-237)
2.In the experiments, it would be better to compare with more related models proposed in recent two years.
Thanks for the comment. We have added two related methods proposed in the recent two years for comparison experiments (TriCNN and HRAM) as suggested. Among them, the TriCNN approach is a three-branch network model based on 3D-CNN using various scale convolutional kernels of 1×1×3, 3×3×1, and 3×3×3 sizes to extract spectral, spatial, and spectral-spatial features respectively, and then fuse the different features obtained from the three sub-networks by feature flattening and concatenation. The HRAM approach employs a hierarchical residual network to extract spatial and spectral characteristics at the granularity level utilizing a two-branch structure in parallel with the corresponding convolution kernel. Besides, to boost the discriminative power of the model, it exploits attention mechanisms to assign adaptive weights to spatial and spectral features at various scales.
In addition, the corresponding comparative experimental results and analysis are described in detail in the “Experimental Results and Discussion” section of the paper. Corresponding content has been marked in red. (see pages 13-21)
3.In the ablation study, it would be interesting to see the performance with only multi-scale 3D DenseNet or only multi-scale 2D Octave.
This is a very good comment indeed. Following the Reviewer’s outstanding suggestion, we have added a description in the section “Ablation Study”. The MCNN denotes that the proposed MOCNN method only adopts multi-scale 3D DenseNet for spectral feature extraction, and there is no branch network employed for spatial information extraction. The MONN denotes that the proposed MOCNN method only adopts multi-scale 2D Octave for feature extraction, and lacks the network for the extraction of spectral information.
From Fig. 12, the classification accuracy of the MONA approach without using any attention mechanism is significantly lower than that of the no-ECA and no-BAM approaches on the four datasets, thus fully proving the effectiveness of the ECA and BAM attention modules. In addition, it’s apparent that the classification outcomes of MONN and MCNN are significantly lower than MOCNN, which is due to the fact that MONN based on multi-scale 2D Octave can only extract spatial feature information at various scales, while lacking access to spectral information. Similarly, MCNN based on multi-scale 3D DenseNet can only withdraw spectral signatures at various scales and lacks the mining of complex spatial information. It also further demonstrates that the single extraction network, either for spectral features alone or for spatial feature information alone, fails to simultaneously sufficiently exploit the spectral-spatial feature information that is favorable for HSI categorization.
Corresponding revised content in the section “Ablation Study” has been marked in red. Thanks again! This is very helpful to improve the quality of my paper. (see pages 21-22, lines 487-501)
4.Some very related works may be considered to cite, such as 'Hyperspectral Image Classification With Attention-Aided CNNs, 2021 TGRS', 'Cross-Modality Contrastive Learning for Hyperspectral Image Classification, 2022 TGRS, 10.1109/TGRS.2022.3188529', 'Multiscale Progressive Segmentation Network for High-Resolution Remote Sensing Imagery, 2022 TGRS, DOI 10.1109/TGRS.2022.3207551'.
This is a very good comment indeed. It has been done in the revised manuscript as suggested. Such as
[Ref 25] Cross-Modality Contrastive Learning for Hyperspectral Image Classification. IEEE Trans. Geosci. Remote Sens. 2022.
[Ref 27] Hyperspectral Image Classification With Attention-Aided CNNs, IEEE Trans. Geosci. Remote Sens. 2021.
5.English language and style are fine/minor spell check required
We apologize for these typos and gratefully thank the Reviewer for his/her careful reading of our manuscript. It has been done in the revised manuscript as suggested. We have gone through the whole manuscript and modified some mistakes.
Last but not least, we would like to take this opportunity to gratefully thank the reviewer again for his/her outstanding comments and suggestions, which greatly helped us to improve the technical quality and presentation of our manuscript.

Reviewer 3 Report
The article is interesting, it includes multi-layered analyses.
Notes on the article:
1. In the article, sections begin and end as a textbook (reasoning and section abstract), and not a presentation of the experimental data obtained. This needs to be changed.
2. It is necessary to adhere to the general structure of the article. Therefore, it is necessary to revise the titles of sections (INTRODUCTION; MATERIALS AND METHODS; RESULTS AND DISCUSSION; CONCLUSIONS; REFERENCES).
3. The output must contain the results, without a description. It is recommended to slightly change the writing style of the section.
Author Response
Responses to comments by Reviewer 3
The article is interesting, it includes multi-layered analyses.
Notes on the article:
We would like to take this opportunity to gratefully thank the reviewer for his/her careful assessment and summary of the main contributions of our work. We also gratefully thank the reviewer for taking the time to review our contribution.
1.In the article, sections begin and end as a textbook (reasoning and section abstract), and not a presentation of the experimental data obtained. This needs to be changed.
This is a very good comment indeed. It has been done in the revised manuscript as suggested. In addition, we have double-checked sections at the beginning and end of the chapters and the introduction sections throughout the paper. Corresponding content has been marked in red. (see pages 2-4, pages 7-9, and page 20)
2.It is necessary to adhere to the general structure of the article. Therefore, it is necessary to revise the titles of sections (INTRODUCTION; MATERIALS AND METHODS; RESULTS AND DISCUSSION; CONCLUSIONS; REFERENCES).
Thanks for the comment.It has been done in the revised manuscript as suggested. In addition, we have changed the titles of the sections (Introduction, Materials and Methods; Experimental Results and Discussion; Conclusions; References) while the descriptions of some of the corresponding content have also been adjusted.
3.The output must contain the results, without a description. It is recommended to slightly change the writing style of the section.
This is a very good suggestionindeed. In the revised manuscript, we have revised this part of the manuscript as suggested, and double-checked the rest of the paper. Corresponding content has been marked in red. (see pages 15-19, lines 372-464) Thank you again for your outstanding advice, this suggestion has greatly helped us to improve the quality of the manuscript.
Last but not least, we would like to take this opportunity to gratefully thank the reviewer again for his/her outstanding comments and suggestions, which greatly helped us to improve the technical quality and presentation of our manuscript.

Round 2
Reviewer 1 Report
Thanks to the authors for the changes. My recommendations have been taking into account and now the paper can be accepted.